

# Contribution of Coastal Retrogressive Thaw Slumps to the Nearshore Organic Carbon budget along the Yukon Coast

Justine L. Ramage[1,2], Anna M. Irrgang[1,2], Anne Morgenstern[1], Hugues Lantuit[1,2]

[1]Department of Periglacial Research, Alfred Wegener Institute Helmholtz Centre for Polar and Marine Research, Potsdam, Germany

[2]University of Potsdam, Institute of Earth and Environmental Science, Potsdam, Germany

*Correspondence to*: Justine L. Ramage (justine.ramage@awi.de)

**Abstract.** We describe the evolution of coastal retrogressive thaw slumps (RTSs) between 1952 and 2011 along the Yukon Coast, Canada, and provide the first estimate of the contribution of RTSs to the nearshore organic carbon budget in this area. We 1) monitor the evolution of RTSs during the periods 1952-1972 and 1972-2011; 2) calculate the volume of material eroded and stocks of organic carbon (OC) mobilized through slumping – including soil organic carbon (SOC) and dissolved organic carbon (DOC) – and 3) measure the OC fluxes mobilized through slumping between 1972 and 2011. We identified
RTSs using high-resolution satellite imagery from 2011 and geocoded aerial photographs from 1952 and 1972. To estimate the volume of eroded material, we applied a spline interpolation on an airborne LiDAR dataset acquired in July 2013. We inferred the stocks of mobilized SOC and DOC from existing related literature. Our results show a 73% increase in the number of RTSs between 1952 and 2011. In the study area, RTSs displaced at least $8600*10^3$ m$^3$ of material, with 53% of ice. We estimated that slumping mobilized $81900*10^3$ kg of SOC and $156*10^3$ kg of DOC. Since 1972, 17% of the RTSs
have displaced $8.6*10^3$ m$^3$/yr of material, with an average OC flux of $82.5*10^3$ kg/yr. This flux represents 0.3% of the OC flux released from coastal retreat; however RTSs have a strong impact on the transformation of OC in the coastal fringe.

## 1 Introduction

Soil organic carbon (SOC) stocks in the Arctic are estimated to 1307 Pg; 76.4% (999 Pg) of them are stored in permafrost terrains (Hugelius et al., 2014). These stocks resulted from slow decomposition of soil organic matter in permanently frozen soils, caused by low soil temperatures and impeded drainage. During the last decades, air temperatures in the Arctic increased by a factor of 3-4, at twice the rate of the global temperature increase (Hansen et al., 2010). As the active layer thickens due to warmer air, increased microbial activity in the soil mobilizes more organic carbon (OC) that is eventually

released to the atmosphere (Mackelprang et al., 2011; Schuur et al., 2008). Organic carbon and nutrients are also released to streams, rivers and to the Arctic Ocean by coastal and riverbank erosion, thermokarst erosion, and thermal erosion (Vonk et al., 2012; Ping et al., 2011; Lamoureux and Lafrenière, 2009). Mass wasting processes along the Arctic coast, such as coastal retrogressive thaw slumps (RTSs), contribute to the transport of terrestrial OC to the nearshore zone (Obu et al., 2016). RTSs are among the most active thermokarst landforms in the Arctic and have increased both in number and size over the past

decades (Segal et al., 2016; Brooker et al., 2014; Lacelle et al., 2010). Active RTS headwalls retreat faster than the coast,





displace a large volume of sediments, and considerably impact the surrounding ecosystems (Schuur et al., 2015; Abbott et al., 2015). RTSs rework sediments and mobilize carbon, nitrogen, and nutrients; as a result RTSs affect terrestrial (Cassidy and Henry, 2016; Tanski et al., 2016; Cray and Pollard 2015; Cannone et al., 2010) and aquatic ecosystems (Malone et al., 2013; Kokelj et al., 2013, 2009a).

Permafrost carbon stocks were recently included in calibrating global carbon models (MacDougall et al., 2012; Burke et al., 2012; von Deimling et al., 2012; Koven et al., 2011; Schaefer et al., 2011). Schaefer et al. (2014) predicted $120 \pm 85$ Gt carbon emissions from thawing permafrost by 2100, which represents $5.7 \pm 4.0\%$ of the total anthropogenic emissions. Nevertheless, global carbon models do not account either for the spatial heterogeneity of permafrost terrains or for abrupt thaw processes (such as thermokarst), post-fire dynamics, or coastal erosion (Hugelius et al., 2014; MacDougall et al., 2012;

Vonk et al., 2012). This gap can be addressed by quantifying the impact of the above processes on the carbon budget (Kuhry et al., 2010).

Our study estimates the impact of thermokarst disturbances on the OC budget in coastal permafrost environments. We calculate the volume of sediments and OC mobilized by the RTSs along the Yukon Coast, Canada. We 1) analyse the evolution of RTSs in the area between 1952 and 2011; 2) calculate the volume of material eroded and stocks of organic

carbon (OC) mobilized through slumping – including soil organic carbon (SOC) and dissolved organic carbon (DOC) – and 3) measure the OC fluxes mobilized through slumping between 1972 and 2011.

## 2 Study area

The study area is located in the Canadian Arctic, along the westernmost coast of the Yukon Territory (Fig. 1). The study area

comprises a 238-km portion of the Yukon Coastal Plain, including Herschel Island (Fig. 1). The area is in the continuous permafrost zone (Rampton, 1982) and tundra vegetation zone dominated by mosses, graminoids, and shrubs (CAVM Team, 2003). The area is characterized by a subarctic climate with mean summer air temperature of 6°C on the East end and 8.7°C on the West end; the mean summer precipitations (June, July and August, 1971-2000) are 79.8 mm on the East end, and 112.9 mm on the West end (Environment Canada, 2017). The Mackenzie River influences seawater temperature and sea ice

extent and is the main forcing on the local precipitation patterns (Burn and Zhang, 2009). The western margin of the Laurentide ice sheet, which reached its maximum ice extent around Herschel Island at ca. 16 200 years BP (Fritz et al., 2012), shaped the topography of the Yukon Coastal Plain. Long and high moraine ridges characterize most of the previously glaciated area. Herschel Island is a moraine thrust at the margin of the formerly glaciated area, and is one of the largest moraine deposits in the region (Mackay, 1959). Stream valleys, fluvial deltas, alluvial fans, and thermokarst basins

characterize the unglaciated area. Due to widespread moraine deposits, 35% of the Yukon Coast is composed of ice-rich cliffs (Harper, 1990). Volumetric ground ice contents (massive ice, pore ice and wedge ice) vary along the coast and range from 0% to 74% (Couture and Pollard, 2017). Previous studies divided the study area into 36 coastal segments (Fig. 1),

based on ground ice contents, surficial geology and geomorphology (Lantuit et al., 2012b; Couture, 2010; Lantuit and Pollard, 2005). Most segments fall into three surficial geologic unit: ice-thrust moraines (30%); lacustrine plains (23%) and rolling moraines (16%). Alluvial fans, stream terraces, floodplains, and outwash plains underlay the remaining segments (Rampton, 1982). The coast is rapidly retreating (Harper, 1990): during the period 1951-2011, the average rate of coastal

5   change was –0.7 m/yr and was characterized by decreasing erosion rates from West to East (Irrgang et al., 2017). RTSs are common along the coast and mostly develop on segments with massive ground ice thicker than 1.5 m and coastal slope greater than 3.9° (Ramage et al., 2017).

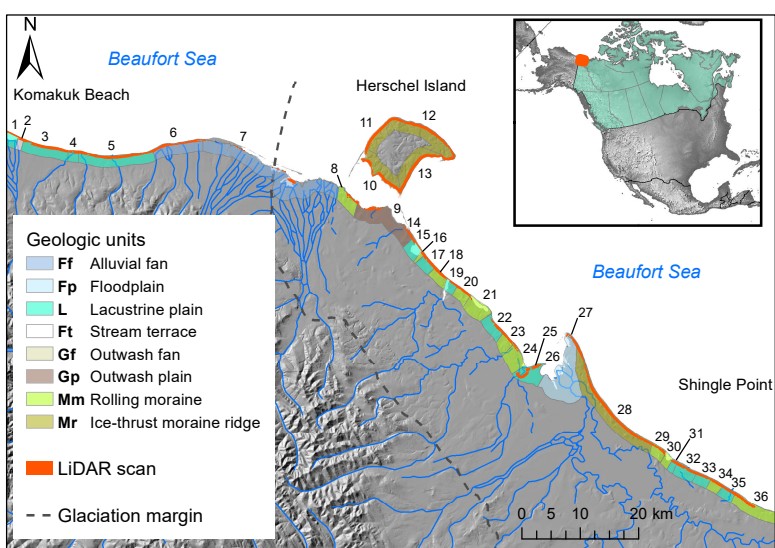

**Figure 1:** Study area. The coastal subset defined as the LiDAR scan is represented in red. The limit of the glaciation was
10   reproduced after Dyke and Prest (1987) and the surficial sediments after Rampton (1982). The numbers stand for the coastal segments stretching along the coast from west to east: 1) Clarence Lagoon West; 2) Clarence Lagoon East; 3) Komakuk Beach West 2; 4) Komakuk Beach West 1; 5) Komakuk Beach; 6) Malcom River Fan; 7) Malcom River Fan with barrier Islands; 8) Workboat Passage West; 9) Workboat Passage East; 10) Herschel Island South; 11) Herschel Island West; 12) Herschel Island North; 13) Herschel Island East; 14) Whale Cove West; 15) Whale Cove; 16) Whale Cove East; 17) Roland
15   Bay northwest; 18) Roland Bay West; 19) Roland Bay East; 20) Stokes Point West; 21) Stokes Point; 22) Stokes Point Southeast; 23) Phillips Bay northwest; 24) Phillips Bay West; 25) Phillips Bay; 26) Babbage River Delta; 27) Kay Point Spit; 28) Kay Point South East; 29) King Point Northwest; 30) King Point Lagoon; 31) King Point; 32) King Point Southeast; 33) Sabine Point West; 34) Sabine Point; 35) Sabine Point East; 36) Shingle Point West.




## 3 Methods

### 3.1 Evolution of RTSs

We used two data inputs to measure the evolution of RTSs between 1952 and 2011: a dataset with RTSs present in 1972 and

5   2011 (dataset A) and a dataset with RTSs present in 1952 (dataset B). All RTSs were mapped using ArcMap 10.3 (ESRI) on

a scale of 1:2000 and classified as active or stable.

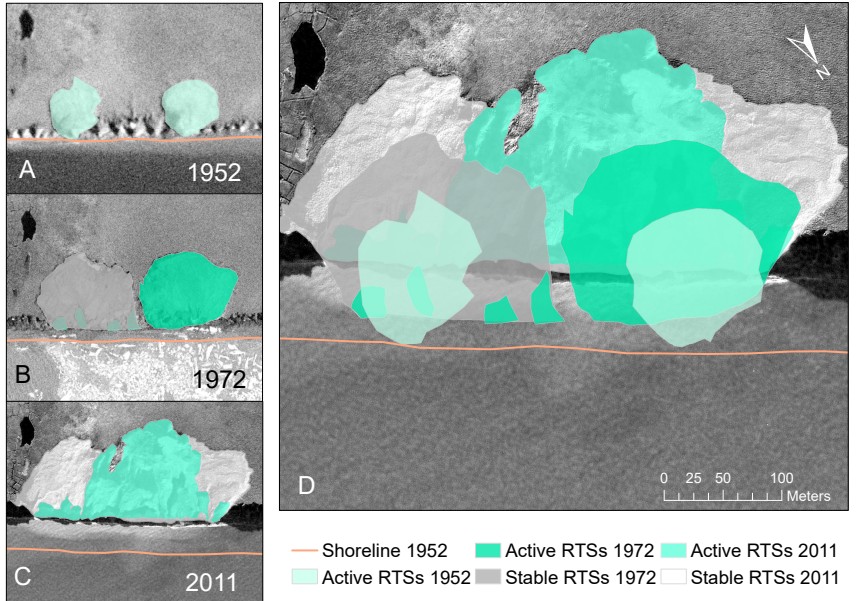

**Figure 2:** Geomorphological map of retrogressive thaw slumps (RTSs) illustrating the complexity of RTS evolution along

10   the Yukon Coast. The underlying imagery is a GeoEye-1 satellite image from 2011 (July 18th). RTSs areas from 1952 and

1972 closer to the shore eroded due to coastal retreat. The remaining parts had either extended and merged with other RTSs

or stabilized in 2011. A) Two active RTSs in 1952 (aerial photo from 1952, National Air Photo Library, Canada). B) RTSs

in 1972 (aerial photo from 1972, National Air Photo Library, Canada). RTSs expanded and one had stabilized. New active

RTSs developed within the stabilized RTS. C) RTSs in 2011 (GeoEye-1, July 18th 2011). Former RTSs had partly stabilized

15   and newer RTSs developed within the boundaries of the stabilized RTSs. D) RTSs present in 1952, in 1972 are overlapping

the 2011 RTSs.



Ramage et al. (2016) provided dataset A. RTSs present in 2011 were mapped based on multispectral GeoEye-1 and WorldView-2 satellite images acquired in July, August and September 2011. RTSs present in 1972 were mapped using a series of geocoded aerial photographs from the 1970s obtained from the National Air Photo Library in Canada (Irrgang et al., 2017). The mapping methodology is explained in detail in Ramage et al. (2017).

Dataset B comprises RTSs present in 1952 that we mapped using a series of geocoded aerial photographs from 1952, obtained from the National Air Photo Library in Canada (Irrgang et al., 2017).

We compared the number and size of RTSs present in 1952, in 1972 and in 2011. RTSs are polycyclic and can occur on surfaces previously affected by RTS. As a result, several active RTSs can be located within the boundary of a stable RTS (Fig. 2). In this case, stable polycyclic RTSs include the areal surfaces of active RTSs located within their boundaries.

### 3.2 Volume estimations

#### 3.2.1 LiDAR dataset

For each RTS identified in 2011 we extracted morphological information – size and mean surface elevation – from an airborne LiDAR dataset acquired in July 2013 (Kohnert et al., 2014). The LiDAR dataset has a scan width of 500 m; the
LiDAR point data was interpolated with inverse distance weighting to obtain digital elevation models with a horizontal resolution of 1 m (Obu et al., 2016). The LiDAR dataset has a final georeferenced point cloud data vertical accuracy of 0.15 ± 0.1 m and covers 80% of the coastline in our study area.

We selected a subset of the 2011 RTSs dataset comprising RTSs that occurred within the boundary of the LiDAR dataset in order to measure the volume of eroded material from RTSs, (Fig. 1). We discarded all RTSs outside of the LiDAR scan from
the volume and flux analyses (n = 125).

Additionally to the RTSs present in 2011 within the LiDAR area, we defined a subgroup with RTSs present in 2011 on surfaces not affected by slumping before 1972; we defined this subgroup as *RTSs initiated after 1972*.

#### 3.2.2 Interpolation method

We applied a regularized spline interpolation technique to model pre-slump topographies used for calculating the volume of
material eroded through slumping. The spline method allows to estimate elevation points outside the range of input sample points and to minimize the total curvature of the surface. We therefore selected spline among other interpolation methods. We based our interpolation on the extensive point elevation data available for the study area from the LiDAR dataset (Fig. 3).



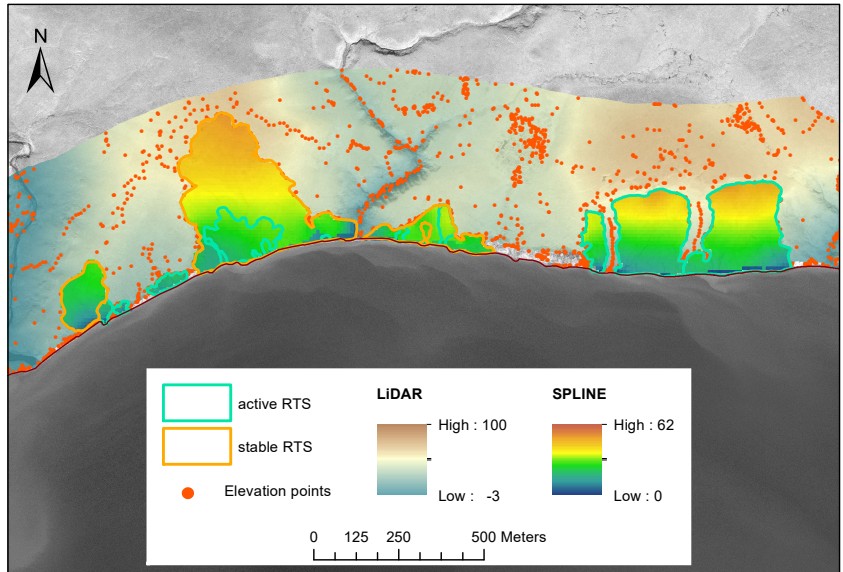

**Figure 3:** Map illustrating the different datasets used to model pre-slump topographies. Retrogressive thaw slumps (RTSs) are outlined in green for the active RTSs and orange for the stable RTSs. The background satellite imagery is a GeoEye-1 image taken on July 18th 2011. The background elevation and the random elevation points outside the RTS areas are derived from the LiDAR dataset. Elevation surface within the RTS borders represent the elevation before RTS occurred and is interpolated using a Spline.

### 3.2.3 Volume of eroded material

In order to calculate the volume of eroded material from the RTS identified in 2011, we subtracted the mean surface elevation values obtained from the LiDAR dataset from the mean interpolated surface elevation values (Fig. 3). Due to ground ice melting, a part of the sediments subside and remain compacted in the RTS floor (Obu et al., 2016). Moreover, coastal retreat erodes the base of RTSs. We did not account for these processes in our analyses.

We derived the volume of eroded ice (VI) and sediments (VS) for each RTS using the volumetric ice content provided in Couture and Pollard (2017). We derived the mass of sediments eroded per RTS using the values provided in Couture (2010).



### 3.3 Estimates of soil and dissolved organic carbon values

We estimated mobilized SOC and DOC stocks and fluxes from RTSs based on the values provided in Couture (2010) and Tanski et al. (2016). OC values were derived from in-situ measurements (Tanski et al., 2016; Couture, 2010) and were

available for each coastal segment.

### 3.3.1 SOC and DOC stocks

We used Equation (1) to estimate the stocks of SOC eroded from RTSs:

$$\textbf{(1)}\ RTS_{SSOC} = \sum_{i=1,\,j=1}^{n,\,m}(M_{CTj} * A_i) + \left(M_{CBj} * \left(VS_i - A_i\right)\right),$$

where $RTS_{SSOC}$ is the stock of SOC eroded from RTSs (expressed in kg); $M_{CTj}$ is the mass of SOC in the upper 1 m

(expressed in kg) per coastal segment $j$ out of $m$ total; $A_i$ is the surface area of an RTS $i$ out of $n$ total (expressed in m$^2$); $M_{CBj}$ is the mass of SOC in the lower soil column (expressed in kg), per coastal segment $j$; and $VS_i$ is the volume of sediment eroded by per RTS (expressed in m$^3$). $M_{CTj}$ and $M_{CBj}$ take into account differences in dry bulk density per coastal segment $j$ (Couture, 2010). We used Equation (2) to estimate the stocks of DOC eroded from RTSs:

$$\textbf{(2)}\ RTS_{SDOC} = \sum_{i=1,\,j=1}^{n,m} D_j * VI_i,$$

where $RTS_{SDOC}$ is the total stock of DOC eroded from RTSs (expressed in kg); $D_j$ is the amount of DOC per coastal segment $j$ (expressed in kg/m$^3$); and $VI_i$ is the volume of ice eroded from a RTS (expressed in m$^3$). $D_j$ is given per coastal segment $j$ (Tanski et al., 2016).

### 3.3.2 SOC and DOC fluxes

We calculated the flux of material – including ice and sediments – as well as SOC and DOC fluxes for the RTSs initiated

after 1972. To calculate the SOC flux we used Equation (3):

$$\textbf{(3)}\ RTS_{FSOC} = RTS_{SSOC} / 39,$$

where $RTS_{FSOC}$ is the annual flux of SOC mobilized from RTSs (expressed in kg/yr); $RTS_{SSOC}$ is the quantity of SOC eroded from an RTS (expressed in kg) (Eq. 1); 39 is the number of years during the time period 1972-2011. Similarly, we used Equation (4) to calculate the DOC flux:

$$\textbf{(4)}\ RTS_{FDOC} = RTS_{SDOC} / 39,$$

where $RTS_{FDOC}$ is the annual flux of DOC eroded from RTSs (expressed in kg/yr); $RTS_{SDOC}$ is the quantity of DOC eroded from an RTS (expressed in kg) (Eq. 2); 39 is the number of years during the time period 1972-2011.



## 4 Results

### 4.1 Evolution of RTSs between 1952 and 2011

#### 4.1.1 RTS evolution along the coast

The number of RTSs increased by 73% between 1952 and 2011. The increase was more pronounced throughout the time
5   period 1952 - 1972 (Table 1). Between 1952 and 2011, active RTSs were more abundant and their number increased faster
than stable RTSs. While the number of active RTSs progressed steadily throughout the period, the number of stable RTSs
decreased between 1972 and 2011 (Table 1): stable RTSs had either reactivated or eroded due to coastal retreat. Between
1952 and 2011, the number of RTSs increased by 40% on lacustrine plains and by 100% on rolling moraines (Table 1). On
ice-thrust moraines, the number of RTSs increased by 69% between 1952 and 2011 (1.2 RTS/yr). On both moraine units, the
10  rise was greater between 1952 and 1972.

**Table 1:** Number of RTSs in 1952, 1972 and 2011 and number of RTSs initiated after 1972, per geologic unit (lacustrine
plains, L; rolling moraines, Mm; ice-thrust moraines, Mr). RTSs initiated after 1972 are a subgroup of RTSs identified in
2011.

|  | RTSs in 1952 | RTSs in 1972 | RTSs in 2011 | RTSs initiated after 1972 |
|---|---|---|---|---|
| **Total** | **166** | **210** | **287** | **119** |
| L | 25 | 27 | 35 | 11 |
| Mm | 42 | 58 | 84 | 20 |
| Mr | 99 | 125 | 167 | 88 |
| Active | 122 | 146 | 203 | 72 |
| L | 21 | 18 | 25 | 5 |
| Mm | 30 | 42 | 53 | 14 |
| Mr | 71 | 86 | 124 | 53 |
| Stable | 44 | 64 | 84 | 47 |
| L | 4 | 9 | 10 | 6 |
| Mm | 12 | 16 | 31 | 6 |
| Mr | 28 | 39 | 43 | 35 |

The total areal coverage (sum of the total RTSs sizes) expanded by 14% between 1952 and 2011 (Table 2) and was observed
in all geologic units. This expansion was driven by an increase in the areal coverage of stable RTSs (25%); the areal
coverage of active RTSs decreased by 2% (Table 2). The expansion in areal coverage was caused by an increase in the
20  number of RTSs rather than by a growth in the size of single RTSs alone: RTSs became smaller, their median size decreased
by 67% throughout the period.



**Table 2:** Median sizes and areal surfaces occupied by RTSs in 1952, 1972 and 2011 and by RTSs initiated after 1972. The data is divided into active and stable RTSs and given per geologic unit (lacustrine plains, L; rolling moraines, Mm; ice-thrust moraines).

| | RTSs in 1952 | | RTSs in 1972 | | RTSs in 2011 | | RTSs initiated after 1972 | |
|---|---|---|---|---|---|---|---|---|
| **Median size (ha)** | **0.75** | | **0.40** | | **0.24** | | **0.16** | |
| L | | 0.19 | | 0.31 | | 0.22 | | 0.14 |
| Mm | | 0.77 | | 0.29 | | 0.17 | | 0.24 |
| Mr | | 1.11 | | 0.47 | | 0.27 | | 0.15 |
| Active | 0.48 | | 0.25 | | 0.15 | | 0.10 | |
| L | | 0.18 | | 0.18 | | 0.17 | | 0.18 |
| Mm | | 0.61 | | 0.24 | | 0.14 | | 0.06 |
| Mr | | 0.59 | | 0.28 | | 0.15 | | 0.11 |
| Stable | 3.50 | | 2.69 | | 1.09 | | 0.68 | |
| L | | 0.99 | | 1.07 | | 1.25 | | 0.98 |
| Mm | | 2.59 | | 2.71 | | 0.44 | | 0.20 |
| Mr | | 4.83 | | 3.39 | | 1.65 | | 0.68 |
| **Total coverage (ha)** | **387.9** | | **384.4** | | **441.9** | | **97.6** | |
| L | | 28.7 | | 28.9 | | 39.7 | | 8.2 |
| Mm | | 87.4 | | 85.6 | | 110.5 | | 3.3 |
| Mr | | 271.8 | | 269.9 | | 291.7 | | 86.1 |
| Active | 162.5 | | 100.3 | | 159.3 | | 16.6 | |
| L | | 24.5 | | 8.6 | | 12.1 | | 1.0 |
| Mm | | 40.2 | | 25.6 | | 27.9 | | 1.1 |
| Mr | | 97.8 | | 66.2 | | 119.3 | | 14.5 |
| Stable | 225.4 | | 284.0 | | 282.6 | | 82.0 | |
| L | | 4.2 | | 20.3 | | 27.6 | | 7.2 |
| Mm | | 47.2 | | 60.0 | | 82.6 | | 2.2 |
| Mr | | 174.0 | | 203.7 | | 172.4 | | 71.6 |

Among RTSs present in 2011, 119 initiated after 1972 on previously undisturbed surfaces: in 2011, 72 were still active and 47 had stabilized (Table 1). RTSs initiated after 1972 were on average smaller than other RTSs (Table 2), and occupied 98.6 ha of the whole study area, or 22% of the total area affected by RTSs in 2011. Most of the RTSs initiated after 1972 (74%) developed on ice-thrust moraines.

### 4.2 Eroded material and estimated amount of mobilized SOC and DOC

In the following sections, volumes are computed for the RTSs that occurred within the LiDAR area. This comprises 56% of the total number of RTSs (n = 162) and 41% of the number of RTSs initiated after 1972 (n = 49).





### 4.2.1 Eroded material and OC stocks mobilized from RTSs

The total volume of material displaced by 162 RTSs was $8600*10^3$ m$^3$, 54% of which was ice (S1, Table S1). On average each RTS eroded $53*10^3$ m$^3$ of material. The volume of eroded material was positively correlated to the size of the RTSs ($r^2$ = 0.5, p < 0.05). Overall, 64% of the material reworked by RTSs originated from ice-thrust moraines, 19% from rolling

5   moraines and 17% from lacustrine plains (Table 3). However, RTSs located on lacustrine plains eroded more material per single RTS ($61*10^3$ m$^3$/RTS) than RTSs located on ice-thrust moraines ($53*10^3$ m$^3$/RTS) and on rolling moraines ($48*10^3$ m$^3$/RTS).

**Table 3:** Volume of material, including ice and sediments, eroded by RTSs along the Yukon Coast per geologic units.

|  | Sediments ($10^6$ m$^3$) | Ice ($10^6$ m$^3$) | Total Material ($10^6$ m$^3$) |
|---|---|---|---|
| Lacustrine Plains | 0.4 | 1.0 | 1.4 |
| Rolling moraines | 0.9 | 0.8 | 1.7 |
| Ice-thrust moraines | 2.7 | 2.8 | 5.5 |

The largest volumes of eroded material came from RTSs occurring at the glaciation limit (Fig. 4). The 24 RTSs located on Herschel Island East (segment 13) reworked 25% of the total volume of material displaced by the 162 RTSs. The RTSs located on Herschel Islands West (segment 11) had the highest volume of material eroded per RTS, on average 2% of the

15   total volume of material displaced by RTSs (Fig. 4). Ice-thrust moraine deposits underlie both coastal segments 11 and 13.





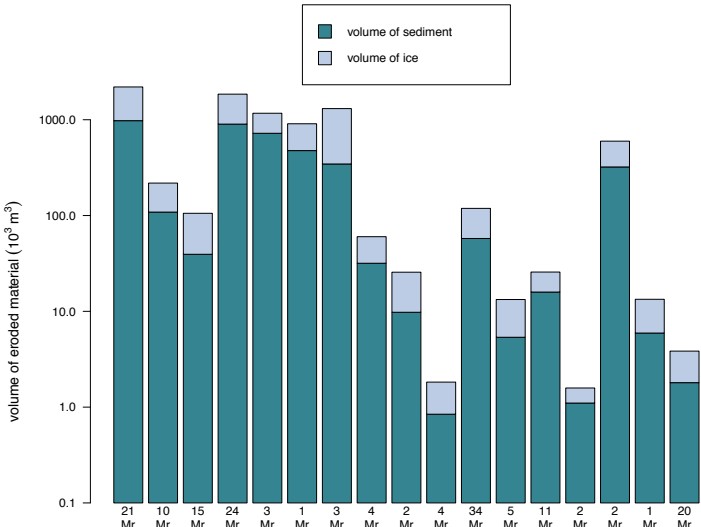

**Figure 4:** Volume of eroded material (sediments and ice) from coastal segments where RTSs occurred in 2011 (logarithmic scale). Each bar corresponds to a coastal segment, following a geographic order from west on the left to east on the right. The values on the x-axis indicate the number of RTSs on the coastal segments. The geologic units are indicated below the bars and referred as L (lacustrine plains); Mm (rolling moraines); and Mr (ice-thrust moraines).

Between 1952 and 2011, the 162 RTSs reworked $8146.6 \times 10^3$ kg of sediments (S1, Table S1), mobilizing a total SOC stock of $81900 \times 10^3$ kg, with the upper 1 m of soil contributing 56%. RTSs on ice-thrust moraines contributed to 69% of the total SOC stock. Out of this, RTSs on Herschel Island West and East (segments 11 and 13) mobilized 47% of the total SOC stock. RTSs on ice-thrust moraines mobilized 65% of the total DOC stock mobilized by the 162 RTSs ($156 \times 10^3$ kg) (S1, Table S1).

### 4.2.2 Eroded material and OC fluxes from RTSs initiated after 1972

The 49 RTSs initiated after 1972 eroded a volume of material of $8.6 \times 10^3$ m$^3$/yr between 1972 and 2011, 50% of which was ice (S1, Table S1). This represents 4% of the total volume of material eroded by the 162 RTSs. The RTS initiated after 1972 on Herschel Island North (segment 12) reworked the largest volume of material: $17 \times 10^3$ m$^3$/RTS (Fig. 5). In total, 95% of the reworked material from RTSs initiated after 1972 came from those located on ice-thrust moraines (Table 4). However, the largest volumes of material per RTS initiated after 1972 came from lacustrine plains (926.5 m$^3$/RTS).

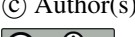



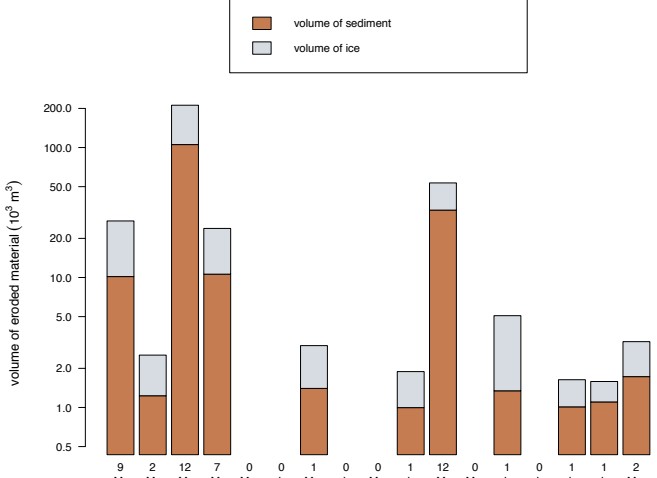

**Figure 5:** Volume of eroded material (sediments and ice) from coastal segments where RTSs initiated after 1972 occurred in 2011 (logarithmic scale). Each bar corresponds to a coastal segment, following a geographic order from West on the left to East on the right. The values on the x-axis indicate the number of RTSs on the coastal segments. The geologic units are

5 indicated below the bars and referred as L (lacustrine plains): Mm (rolling moraines): and Mr (ice-thrust moraines).

**Table 4:** Volume of material, including ice and sediments, eroded by RTSs initiated after 1972 along the Yukon Coast per geologic unit.

|  | Sediments ($10^3 \text{ m}^3$) | Ice ($10^3 \text{ m}^3$) | Total Material ($10^3 \text{ m}^3$) |
|---|---|---|---|
| Lacustrine Plains | 4.4 | 5.7 | 10.1 |
| Rolling moraines | 3.1 | 3.0 | 6.1 |
| Ice-thrust moraines | 160.4 | 158.3 | 318.7 |

The 49 RTSs initiated after 1972 reworked $341.2 \times 10^3$ kg/yr of material and therefore mobilized an SOC flux of $82.3 \times 10^3$ kg/yr (Table 5), representing an average of 10.4 kg/m$^3$/yr. Most of the SOC fluxes originated from the RTSs initiated after 1972 on Herschel Island North (segment 12, $43.2 \times 10^3$ kg/yr) and on Kay Point South East (segment 28, $18.9 \times 10^3$ kg/yr) (S1, Table S1). On ice-thrust moraines, RTSs initiated after 1972 mobilized 92% of the total SOC flux (Table 5). The total DOC

15 flux from RTSs initiated after 1972 was 182.6 kg/yr, with high variability between the coastal segments: from 0.6 kg/yr to





122.7 kg/yr (S1, Table S1). The highest DOC fluxes came from ice-thrust moraines from Herschel Island North (segment 12) where 12 RTSs initiated after 1972 mobilized a total flux of 122.7 kg/yr of DOC (Table 5).

**Table 5:** Total SOC and DOC flux mobilized between 1972 and 2011 by RTSs initiated after 1972, per geologic unit
(lacustrine plains, L; rolling moraines, Mm; ice-thrust moraines, Mr).

|  | SOC flux ($10^3$ kg / yr) | DOC flux (kg / yr) |
|---|---|---|
| L | 3.6 | 4.3 |
| Mm | 3.4 | 3.1 |
| Mr | 75.3 | 175.2 |
| Total | 82.3 | 182.6 |

## 5 Discussion

### 10 5.1 Acceleration of slump activity

The number of RTSs along the Yukon Coast increased by 73% between 1952 and 2011, when on average 2 RTSs initiated per year. The rise was more pronounced between 1952 and 1972 and the number of RTSs continued to increase steadily between 1972 and 2011. The evolution of RTSs along the Yukon Coast is consistent with the observations made in other parts of the Canadian Arctic, where RTS activity is accelerating since the 1950s (Segal et al., 2016; Lacelle et al., 2010;
Lantz and Kokelj, 2008; Lantuit and Pollard, 2008). Lantuit and Pollard (2008) showed that the number of RTSs on Herschel Island increased by 61% between 1952 and 2000. RTSs develop following changes that affect geomorphic settings (Ramage et al., 2017; Kokelj et al., 2017) and are induced by climatic conditions – such as increased air temperature (Lacelle et al., 2010), precipitation events (Kokelj et al., 2015; Lacelle et al., 2010) and storm events (Lantuit et al., 2012a; Lantuit and Pollard, 2008; Dallimore et al., 1996). Many RTSs that were stable or stabilized between 1952 and 1972 re-activated
between 1972 and 2011. Our results confirm the pattern of RTS reactivation previously observed on Herschel Island (Lantuit and Pollard, 2008) and between Kay Point and Shingle Point (Wolfe and Dallimore, 2001) and referred to as polycyclicity. Reactivation of RTSs is associated with incomplete melting of massive ice during the first period of RTS development (Burn, 2000) and depends on the capacity of the slump headwall to remain exposed until ice is exhausted. In coastal settings, storm events can re-activate RTSs (Lantuit et al., 2012a). The period of RTS activity partly depends on the equilibrium
between thermodenudation and coastal erosion rates: the RTS remains active if the RTS headwall erodes at a rate exceeding





coastal retreat (Lantuit et al., 2012a; Are, 1999). This equilibrium is strongly linked to the changing climatic and sea conditions that act on coastal retreat, such as storm events and sea ice duration.

Along the Yukon Coast, RTSs developed mainly on ice-thrust moraines, where their number increased by 1.1 RTS/yr throughout the whole period. Differences in ice content and coastal geomorphology explain the disparities in the evolution of RTSs observed among geologic units (Ramage et al., 2017; Lewkowicz, 1987). Our results confirm the results of Kokelj et al. (2017), who showed evidence of a spatial link between RTS occurrence in North America and the maximum extent of the Laurentide Ice Sheet. Similar to our observations along the Yukon Coast, most of the RTSs in North America are found along the marginal moraines of the Laurentide Ice Sheet.

Along with the increase in number of RTSs along the Yukon Coast, the total areal coverage of RTSs increased by 14% between 1952 and 2011. However, RTSs along the Yukon Coast were on average smaller in 2011 compared to 1952 and 1972. This differs from RTSs observed in other parts of the Canadian Arctic (Segal et al., 2016; Kokelj et al., 2017). Our results support those reported by Ramage et al. (2017); coastal RTSs are on average smaller compared to inland RTSs, and coastal RTSs along the Yukon Coast are smaller than the ones found in other coastal areas of the Arctic. The large number of RTSs initiated after 1972 along the Yukon Coast partly explains this: RTSs initiated after 1972 represented 41% of the total amount of RTSs in 2011; these RTSs were still developing in 2011 and thus did not reach their maximal expansion size.

### 5.2 Eroded material from RTSs

According to our estimates, RTSs have reworked at least $8600*10^3$ m$^3$ of material along the Yukon Coast, among which $335*10^3$ m$^3$ of material was reworked by 49 RTSs initiated after 1972. These estimates are low compared to material removal from other RTSs in the Arctic. Lantuit and Pollard (2005) calculated a sediment volume loss of $105*10^3$ m$^3$ between 1970 and 2004 for a single RTS located on Herschel Island; Kokelj et al. (2015) and Jensen et al. (2014) measured material displacements up to $10^6$ m$^3$ per RTS located in NW Canada and Alaska; the Batagay mega-slump located in Siberia eroded more than $24*10^6$ m$^3$ of ice rich permafrost in 2014 (Günther et al., 2015). The size of the observed RTSs is one reason behind such differences: most of the RTSs examined in the above studies are classified as mega slumps (> 0.5 ha). RTSs along the Yukon coast are small, with an average size of 0.2 ha (Ramage et al., 2017). Furthermore, our estimates do not include all of the material eroded by the RTSs; they only represent the amount of material released to the nearshore zone through slumping. We did not include the material eroded from the RTS headwalls that settles within the RTS floors and the material eroded and transported alongshore by coastal erosion (Fig. 6).





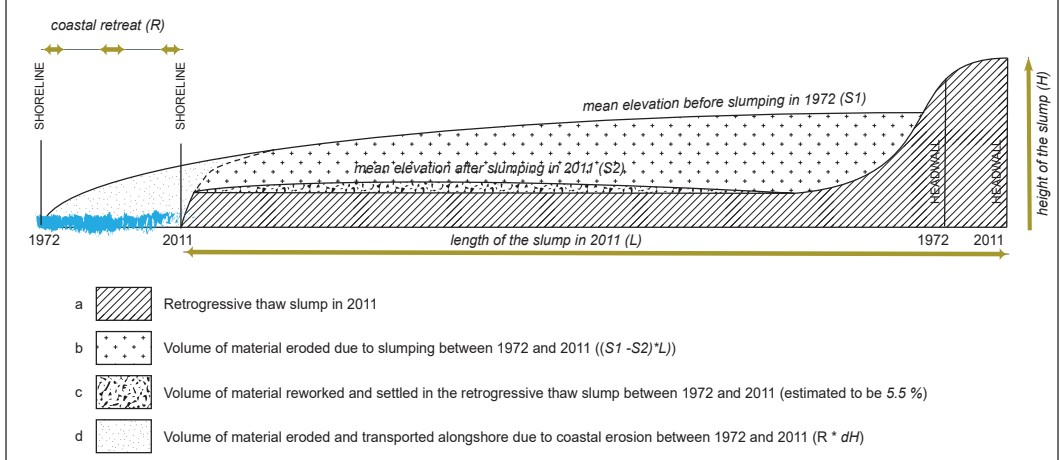

Figure 6: Cross-section of a retrogressive thaw slump (RTS) illustrating the calculated and omitted volumes of sediments eroded through slumping between 1972 and 2011. The calculation estimates the amount of material released to the nearshore zone through slumping (b). It does not take into account the material eroded from the RTS headwalls that remains within the

RTS floors where it settles (c), and (d) the material eroded and transported alongshore by coastal erosion.

On average, 5.5% of the material eroded from RTSs within a year is temporarily accumulated in RTS floors (Obu et al., 2016). There, melted ground ice and sediments are mixed and water transports most of the sediments to the nearshore zone. Without enough viscous flow, the remaining part of the sediments accumulates within the RTS floor and settles. The

settlement is noticeable through the higher bulk densities in samples from RTS floors (Tanski et al., 2017; Lantuit et al., 2012a). Another part of the material mobilized through slumping is transported alongshore by coastal erosion (Fig. 6). In our study area, RTSs incised 8.3% (15.8 km) of the coastline in 2011 and the average rate of coastal change is -0.7 m/yr (Irrgang et al., 2017). Couture (2010) estimated to $7.3*10^6$ kg/km the annual flux of material eroded by coastal retreat along the Yukon Coast. Scaled to the 190-km LiDAR scan in this study, the annual flux of eroded material is $1387*10^6$ kg/yr. We

estimate that RTSs contribute to 8.3% ($111*10^6$ kg) of the annual material budget eroded from coastal erosion along the Yukon Coast.

### 5.3 Calculated OC fluxes

We estimated the annual OC fluxes (SOC and DOC) from 49 RTSs initiated after 1972 to $82.5*10^3$ kg/yr. The average OC

flux from coastal retreat along the entire Yukon Coast is $157*10^3$ kg/km/yr (Couture, 2010) with an annual average DOC



flux of $0.2*10^3$ kg/km/yr (Tanski et al., 2016). In the 190-km coastline of the LiDAR scan, the OC flux from coastal retreat is therefore $29830*10^3$ kg/yr (including $38*10^3$ kg/yr of DOC).

The annual OC flux released by the 49 RTSs initiated after 1972 was 0.3% the annual OC flux from coastal retreat (0.5% of the DOC flux). Most of the OC fluxes from RTSs originated from ice thrust moraines, where the number of RTS initiated

after 1972 was the highest. RTSs develop mainly on ice-thrust moraines because of the presence of large volumes of massive ground ice (Ramage et al., 2017). As a result, only half of the material eroding from the RTS headwall is sediment.

Our fluxes of sediment and OC mobilized by 49 RTSs initiated after 1972 underestimate the annual contribution from slumping processes to the nearshore sediment and OC budgets along the Yukon Coast. These fluxes account for 41% of the

flux from RTSs initiated after 1972 and 17% of the total number of RTSs present in 2011 along the Yukon Coast (Ramage et al., 2017).

### 5.3 Impact of RTSs on the coastal ecosystem

RTSs erode surfaces and leave scars on the landscape, impacting the coastal fringe ecosystems. RTSs alter vegetation

composition and their effects on the vegetation persist over centuries even after stabilization of the RTSs (Cray and Pollard, 2015; Lantz et al., 2009). Similarly, RTSs also play an important role for the coastal ecosystem because they mobilize carbon prior to its release to the ocean and modify the amount of carbon available for the coastal ecosystem (Tanski et al., 2017; Cassidy and Henry, 2016; Pizano et al., 2014). Tanski et al. (2017) show that TOC and DOC decrease by 77% and 55% before reaching the nearshore zone and most of the degradation and mineralization take place within RTS floors.

Abbott et al. (2015) describe similar processes for RTSs located in upland areas, where they observed the removal of 51% of organic-layer SOC and an average loss of 21 kg/m$^2$ mineral-layer SOC following the development of RTSs. Part of this carbon is released to the atmosphere as $CO_2$ (Cassidy and Henry, 2016), another part is buried and accumulates in the RTS floor and the rest is transported through streams and eventually reaches the nearshore zone (Tanski et al., 2017). Kokelj et al. (2013) showed that RTSs strongly impact stream sediment transport by raising the stream turbidity and the concentration of

total suspended sediments.

Similarly, RTSs impact coastal processes by altering coastal retreat (Obu et al., 2016; Lantuit and Pollard, 2008; Leibman et al., 2008). Spatial variability of coastal erosion along the Yukon Coast is partly related to RTS activity: sections of the coast affected by intense slumping activity have the strongest volumetric values of erosion and accumulation (Obu et al., 2016).

The impact of RTSs on the ecosystem is important even though RTSs are transient phenomena in coastal setting. The

material reworked by RTSs is not an additional contribution to the coastal budget, but a part of the cliff material that is eventually eroded by coastal retreat. While the OC mobilized through cliff erosion is transported in the nearshore zone (Vonk et al., 2012) a fraction (5.5%) of the OC mobilized through slumping remains for several years in the slump floor (Tanski et al., 2017; Obu et al., 2016), where it is mineralized by microorganisms.



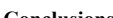

**Conclusions**

The number of RTSs along the Yukon Coast increased by 73% between 1952 and 2011. We observed disparities between geomorphic units: the largest increase was on ice-thrust moraines, where the number of RTSs increased at an annual rate of 1.2 RTSs/yr. Many RTSs are polycyclic and reactivated between 1972 and 2011. RTSs reworked at least $8600*10^3$ m$^3$ of

material within a 190-km portion of the coastal fringe. The OC flux from 49 RTS initiated after 1972 and present in 2011 was $82.5*10^3$ kg/yr and represented 0.3% of the annual OC fluxes from coastal erosion. This number accounts for 17% of the RTSs present along the Yukon Coast in 2011. Our results do not include the volume of material eroded from the RTS headwalls (that remained within the RTS floors where it subsided) and material eroded from the RTS bluff by coastal retreat and transported alongshore. However, we provide a first estimate on the contribution of RTSs to the nearshore carbon budget

along the Yukon Coast.

*Author contribution:* JLR and HL designed the study. AMI geocoded the historical photographs used for mapping. JLR created the spline interpolation and calculated the eroded volumes of material from retrogressive thaw slumps. JLR prepared the manuscript with contributions from all co-authors.

*Aknowledgments:* This study was supported by the Helmholtz Association through the COPER Young Investigators Group

(VH-NG-801) and by the Alfred Wegener Institute in Potsdam. J. L. Ramage was financially supported by a PhD stipend by the University of Potsdam. A.M. Irrgang was financially supported by a PhD stipend from the German Federal Environmental Foundation. We thank J. Obu and G. Hugelius who provided insight and expertise at the early stage of the study.

*Competing interests*: The authors declare that they have no conflict of interest.



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

5    Shingle Point to Kay Point, Yukon Territory, *Rep.*, Geological Survey of Canada