# Peer review of "Increasing coastal slump activity impacts the release of sediment and organic carbon into the Arctic Ocean"

_Biogeosciences, 2017_

## Referee Comment (RC1) · Anonymous Referee #1 · 25 Nov 2017

This paper entitled, "Contribution of coastal retrogressive thaw slumps to the nearshore organic carbon budget along the Yukon Coast," by Ramage and others uses repeat analysis of satellite and LiDAR imagery to assess the number, area, and volume of retrogressive thaw slumps. They found that the number of slumps increased from 1952-2011, but the area affected by slumps changed little. Slumps displaced a large volume of soil and dissolved organic carbon. This study produces an data set htat is very relevant to an important source of uncertainty in understanding how permafrost landscapes and the organic matter they contain are responding to climate change: thermo-erosion. This process has proven difficult to model and the geophysical and ecological consequences of thermos-erosion on landscape and regional scales remain

uncertain. I have a few questions and comments about the methodology, but my main concern is that the current paper quickly gets into the details of these sites and then remains largely descriptive and stops short of positioning these findings in a broader ecological/landscape perspective. If revised with a broader focus, I think this paper would be a valuable contribution to this journal and the larger discourse on the effects of thermo-erosion features on permafrost landscape evolution during climate change. I outline my main questions and concerns below, followed by line edits:

1. This study presents valuable data that are difficult to acquire about the extent and volume of sediment affected by thermo-erosion on decadal timescales. However, I felt it did not fully exploit these data, remaining largely observational and not providing a clear discussion of how these data relate to larger questions about ecosystem carbon balance, links between geomorphology and climate, and permafrost ecology. Given the spatial and temporal richness of this data set, in addition to describing the changes in thermo-erosion area and volume, are there underlying mechanisms the authors could explore? For example, do differences in precipitation, aspect, or other parameters affect rate of thermo-erosion? How representative is this area compared to other Arctic coasts? How different were changes in air temperature for the two periods and is this associated with changes in thermo-erosion? How much of the slowdown in feature formation is due to depletion of ground ice versus external forcing?

2. At the end of the study, I was left wondering what the conclusions were in relation to the core questions/purposes of the study (how is thermo-erosion changing through time). Clearer statement of the purpose of the study would help this, as currently the results quickly get into comparisons within the dataset (e.g. % of sediment reworked done by an individual feature), leaving me confused as to whether thermo-erosion is expanding in this area and if formation is accelerating. The issue of units (addressed below) compounded this confusion.

3. I found the units of sediment and carbon counterintuitive and difficult to compare with other studies. Results are presented in absolute terms (total amount of carbon or

sediment displaced from the whole study region) and it would be useful to state units normalized to area. Expressing material balance in terms of m2 would immediately let researchers unfamiliar with this area relate to the units and assess how important this process is. That would allow comparison of thermokarst mobilization of SOC and DOC to carbon release via active layer deepening. In this same vein, the number of features, which is focused on in the abstract and throughout the paper, seems immaterial compared to changes in area and volume. Ultimately, I had a hard time concluding at the end of the paper if thermo-erosion was increasing, decreasing, or remaining stable.

4. It is unclear how/if uncertainties were propagated through this exercise. Absolute numbers are given, rather than ranges or estimates of center and standard deviation (e.g. all the tables and figures). Without measures of uncertainty, it is difficult to assess the reliability of these estimates or identify sources of that uncertainty in the analysis.

5. There are multiple issues with visualizations—particularly the stacked bar plots using a logarithmic y-axis and the reliance on tables. Stacked bar plots on a logarithmic scale are visually misleading since the ice volume, which represents the majority of material lost, appears negligible. Additionally, could the x-axis of these plots be organized by some salient ecological parameter (e.g. precipitation, climate, surficial geology) instead of by geographic position? This would help provide insight into processes driving these patterns. The use of tables is fine in some cases, but I wanted a figure showing rate of thermo-erosion (normalized by area) for the two time periods (1952-1972, 1972-2011), which seems like one of the key punchlines of this paper. The tabular form makes it harder to rapidly compare changes and trends and ultimately is not more compact than a (non-logarithmic) stacked barplot of those time periods.

6. To cryosphere scientists, the subject of this paper is immediately of interest, but I fear that the abstract and introduction do not provide enough context for a non-specialist to see the need and implications of the study. Defining key terms (e.g. active layer) and providing more context for why this process is of general interest would increase the impact of this paper.

7. The paper builds on many previous studies, but sometimes relies too heavily on explanations given in those studies. Especially on key issues like determining preformation ice content, DOC, and SOC, enough methodological detail should be given for the reader to assess the approach. At the bare minimum, given that many of these estimates are highly uncertain (e.g. reconstructions of ice content), an explicit treatment of uncertainties and how uncertainties were propagated is necessary.

Line edits: Page 1 Line 10: An additional line introducing the general context would be valuable Line 17-18: Standard SI format for number should be used (i.e. $8.6 \times 10^6$ not $8600 \times 10^3$). There are issues with this throughout the manuscript. Line 18: 53% of which was ice Line 21: 0.3% of the total OC flux for the Arctic Ocean? Unclear why this is of interest at this point in the paper. What percentage of the SOC stocks in the affected areas of the study region was mobilized by these features? Line 25: I believe this estimate is for the entire permafrost zone, not just the Arctic Line 27: Is it meant that air temperature has increased by approximately 3-4 degrees C? Air temperature in Celsius is expressed on a relative scale and it does not make sense to say increased by a factor of 3-4 (unless referring to change relative to absolute zero) Line 31: Non-standard terminology for thermo-erosion features. Following Kokelj, Jorgenson, Fortier etc., thermo-erosion or thermal erosion are the blanket terms that include thermokarst (permafrost collapse) and other erosive processes associated with permafrost degradation.

Page 2 Line 5: Consider including more recent modeling studies such as Koven et al 2015, Kessler 2017, or Sudakov and Vakulenko Line 9: Consider citing Abbott et al. 2016 or McGuire et al 2016, which summarize current modeling uncertainties stemming from exclusion of these parameters. Both of these studies directly support the need for the current study by emphasizing the importance of constraining thermo-erosion. Line 25: Word choice (potentially control or influence rather than forcing) Figure 1: Really nice figure. Potentially put the specific reach names in the SI (not of interest to most readers)

Page 6 Line 9: "In order to" can always be replaced by "To" Line 8: How was uncertainty for the compound assumptions in these analyses dealt with? Need more detail generally. Line 12: Why were these processes not included? How does that affect the estimates?

Page 7 Line 6-26: With the presented information, it is not clear if these estimates were downscaled from measurements of fluxes at feature outlets or if they are inferred from the mass of SOC there previously multiplied by volume displaced. If the latter, how are vertical differences in SOC accounted for this this framework?

Page 8 Line 3: Focusing on the number of features doesn't seem terribly relevant to the question of the permafrost climate feedback. The area and volume results are more informative. In general, a few clear figures would more effectively communicate the observed patterns. Table 1: This would be more compelling in figure form. If table is retained, no need to use cryptic acronyms in the first column (i.e. L, Mm, Mr)—there is enough room to spell out the parameters

Page 9 Table 2 would also be more effective in figure format. As currently presented, it is hard to tease apart what is changing across the timeseries.

Page 10 Table 3: This should be normalized to area covered by the geologic units. Are some of the units displacing more material per unit area or are the differences due to different relative coverages? No estimates of uncertainty are given. Figures 4 and 5. Problematic to show a stacked bar plot with a logarithmic axis.

Page 14 Line 18: Good example of why estimates should be normalized by area (i.e. expressed on a kg m2 yr basis or in mm/yr for sediment) Line 27: Still not clear how this affects the analysis. If the question is about total sediment and carbon balance, these processes, which are caused or at least facilitated by thermo-erosion seem pertinent

Page 16 Line 20: I think the authors are referring to Abbott and Jones 2015

References: Abbott, B.W. & Jones, J.B. (2015). Permafrost collapse alters soil carbon

stocks, respiration, CH4, and N2O in upland tundra. Glob. Change Biol., 21, 4570–4587.

Abbott, B.W., Jones, J.B., Schuur, E.A.G., III, F.S.C., Bowden, W.B., Bret-Harte, M.S., et al. (2016). Biomass offsets little or none of permafrost carbon release from soils, streams, and wildfire: an expert assessment. Environ. Res. Lett., 11, 034014.

Kessler, L. (2017). Estimating the economic impact of the permafrost carbon feedback. Clim. Change Econ., 08, 1750008.

Koven, C.D., Schuur, E. a. G., Schädel, C., Bohn, T.J., Burke, E.J., Chen, G., et al. (2015). A simplified, data-constrained approach to estimate the permafrost carbon–climate feedback. Phil Trans R Soc A, 373, 20140423.

McGuire, A.D., Koven, C., Lawrence, D.M., Clein, J.S., Xia, J., Beer, C., et al. (2016). Variability in the sensitivity among model simulations of permafrost and carbon dynamics in the permafrost region between 1960 and 2009. Glob. Biogeochem. Cycles, 30, 2016GB005405.

Sudakov, I. & Vakulenko, S.A. (2015). A mathematical model for a positive permafrost carbon–climate feedback. IMA J. Appl. Math., 80, 811–824.

---

## Referee Comment (RC2) · Anonymous Referee #2 · 26 Nov 2017

The submitted paper 'Contribution of Coastal Retrogressive Thaw Slumps to the Nearshore Organic Carbon Budget along the Yukon Coast' by Ramage et al. gives an indication of the specific impacts of slumps on the sediment budget and on the carbon budgets of Arctic tundra coasts in northern Canada. The focus is on three main topics as stated at the end of the introduction: 1) definition, quantification and temporal analysis of RTSs; 2) estimation of sediment/ice and OC budgets related to these slumps; and 3) measure the OC fluxes between 1972 and 2011. Looking to these aims of the study, I have some specific comments related to these different goals and will come with suggestions to restructure some parts of the paper to make it more focused on the RTSs. The data presented is very valuable and the paper will after restructur-

ing be a valuable contribution to better understand Arctic coastal environments and its changes.

Ad 1). The coastal stretch in NW Canada is probably very representative for a large part of the Arctic coastal environments. The different geomorphological units (or geological units as stated in Figure 1) cover a wide range of environments and the coastal stretch with its units can probably be used to upscale the findings. You can state explicitly that you use the findings along this stretch to upscale in the future. Use Figures 2 and 6 to define what RTSs are. Tables 1 and 2 give an indication of the amount and sizes of RTSs for different units. It would be good to start the explanation of the spatial pattern, followed by the development in time. Now, it starts with changes in time, without having an idea how many and how large the RTSs are in the different units. Another thing is the use of the terms active or stable RTS. What kind of conditions do you use to call a RTS active or stable? Is it related to fresh scarps, vegetation coverage?

Ad 2). Estimation of the sediment/ice erosion due to RTS and OC budgets is quite straightforward and the best you can do with the limited Lidar data. Figure 6 in your discussion is very nice and can be used to explain your estimation of budgets in an earlier phase. The presentation in Figures 4 and 5 is often a bit confusing. You followed a spatial axis on the x-axis, but you don't use this in the rest of the discussion. It would probably more interesting to group it according to the geomorphologic/geologic unit and discuss this variation in time. You also gave this unit-related results in tables 3,4 and 5. The volumes of eroded materials will be better visible if you don't use a cumulative sediment and ice volume on a logarithmic Y axis, but come with values for sediment and ice (different symbols).

Ad 3). I wonder why you made a separate aim only following OC budgets in time. I think it is more logical to describe the estimation of budget terms under 2) and therafter discuss all terms (sediment / ice / OC) in more detail in time.

Other general comments:

Discussion and conclusions The discussion of the paper is now in 4 subsections (erroneous numbered 5.1, 5.2, 5.3 and 5.3). Following the three aims and the structure of the results, it is perhaps very attractive to start a discussion about the 'static' description of RTSs (sizes, amounts, coupling to geo units) and about the uncertainties in determining the RTSs using the data set. This can be followed by a second section about the changes in slump activity (your acceleration of slump activity). Then we have two sections related to the second aim: your sections about Eroded material from RTSs and Calculated OC fluxes. Finally, you can place it in a broader prespective as you tried to do in 5.4. This can also include some remarks about upscaling to Arctic shorelines. The conclusions are to the point.

Title The title is not covering the work done. You have showed many more results on the losses of ice and sediments and its changes in time as well. Impact not only OC budgets.

Small things: Page 2, line 9: Hugelius et al., 2014 is not in the reference list. Page 5, line 20: n=125 refers to? Page 6, Figure 3: Are all splines in the RTSs giving a sloping surface from N to S? Page 6, line 12: Can you estimate the coastal retreat impact (or assume ..%)? Page 11, Figure 4: Only Mr? Page 13, line 21: Wolfe and Dallimore or Wolfe et al. (see reference list) Page 16, line 13: Should be section 5.4.

---

## Author Comment (AC1) · 18 Jan 2018

Thank you for the thorough revision of our manuscript and your constructive comments that helped to improve the paper. Our replies to your comments are combined in the .pdf "Answer_to_reviewers_JR". We added the 3 figures that we modified (Figures 5,6, and 7).

Further modifications can be found in the reviewed version of the manuscript.

Best, Justine Ramage

Please also note the supplement to this comment:
https://www.biogeosciences-discuss.net/bg-2017-437/bg-2017-437-AC1-supplement.pdf

[Figure]

**Fig. 1.** Figure 5

[Figure]

**Fig. 2.** Figure 6

[Figure]

**Fig. 3.** Figure 7

**Supplement:**

*We thank the two reviewers and the editor for the thorough revision of our manuscript and their constructive comments that helped to improve the paper. Our replies to the comments are written in green. Line numbers given in our replies refer to the revised version of the manuscript.*

**Anonymous Referee #1**

This paper entitled, "Contribution of coastal retrogressive thaw slumps to the nearshore organic carbon budget along the Yukon Coast," by Ramage and others uses repeat analysis of satellite and LiDAR imagery to assess the number, area, and volume of retrogressive thaw slumps. They found that the number of slumps increased from 1952-2011, but the area affected by slumps changed little. Slumps displaced a large volume of soil and dissolved organic carbon. This study produces an data set that is very relevant to an important source of uncertainty in understanding how permafrost landscapes and the organic matter they contain are responding to climate change: thermo-erosion. This process has proven difficult to model and the geophysical and ecological consequences of thermos-erosion on landscape and regional scales remain uncertain.

I have a few questions and comments about the methodology, but my main concern is that the current paper quickly gets into the details of these sites and then remains largely descriptive and stops short of positioning these findings in a broader ecological/landscape perspective. If revised with a broader focus, I think this paper would be a valuable contribution to this journal and the larger discourse on the effects of thermo-erosion features on permafrost landscape evolution during climate change. I outline my main questions and concerns below, followed by line edits:

1. This study presents valuable data that are difficult to acquire about the extent and volume of sediment affected by thermo-erosion on decadal timescales. However, I felt it did not fully exploit these data, remaining largely observational and not providing a clear discussion of how these data relate to larger questions about ecosystem carbon balance, links between geomorphology and climate, and permafrost ecology. Given the spatial and temporal richness of this data set, in addition to describing the changes in thermo-erosion area and volume, are there underlying mechanisms the authors could explore? For example, do differences in precipitation, aspect, or other parameters affect rate of thermo-erosion? How representative is this area compared to other Arctic coasts? How different were changes in air temperature for the two periods and is this associated with changes in thermo-erosion? How much of the slowdown in feature formation is due to depletion of ground ice versus external forcing?

   We agree, there is a need for a better understanding of the role of these processes on the development of retrogressive thaw slumps along the Yukon Coast (RTSs). However, this goes beyond the scope of our manuscript and requires a publication on its own.
   However, we updated the manuscript to draw the attention on these issues in the section 5.1 of the revised manuscript. Based on the climate records provided by Environment Canada, we looked at the change in the mean air temperature and average precipitations for the periods 1957-1971 and 1971-2000. Based on these data we could show that:
   **Page 14, line 27**: "*Climate data recorded at Komakuk Beach (segment 2) and Shingle Point (segment 36) show that the average summer air temperature decreased between the periods 1957-1971 (Komakuk, 7.4°C; Shingle Point, 10.8°C) and 1971-2000 (Komakuk, 4.9°C; Shingle Point, 7.4°C). However, the annual average precipitation increased at both stations by 30% and 41%, respectively during the same periods (Environment Canada, http://climate.weather.gc.ca/historical_data/search_historic_data_e.html). Similar patterns were observed for the summer months (July to September). As suggested by Kokelj et al.*

*(2015) in other Arctic areas, higher rainfall might intensify RTS activity. However, a series of environmental factors seems to be jointly responsible for the intensification of RTS activity along the Yukon Coast (Ramage et al., 2017)."*

2. At the end of the study, I was left wondering what the conclusions were in relation to the core questions/purposes of the study (how is thermo-erosion changing through time). Clearer statement of the purpose of the study would help this, as currently the results quickly get into comparisons within the dataset (e.g. % of sediment reworked done by an individual feature), leaving me confused as to whether thermo-erosion is expanding in this area and if formation is accelerating. The issue of units (addressed below) compounded this confusion.

We modified the conclusion to remove any confusion and to make our statement clear: RTSs have a non-negligible impact on the nearshore zone.
**Page 17, line 17:** *"The number of RTSs along the Yukon Coast increased by 73% between 1952 and 2011 and the total areal coverage of RTSs increased by 14%. We observed disparities between geomorphic units: the largest increase was on ice-thrust moraines, where the number of RTSs increased at an annual rate of 1.2 RTSs/yr. Many RTSs are polycyclic and reactivated between 1972 and 2011. RTSs reworked at least 16.6\*106 m3 of material within a 190-km portion of the coastal fringe. Majority of the material came from erosion of the headwall (53%) and 3% remained in the RTS floors. A large amount of the material from RTSs was eroded and transported alongshore due to coastal retreat (45%). The OC flux from 17% of the RTSs identified in 2011 was 1.3\*103 kg/km/yr and represented 0.6% of the annual OC fluxes from coastal retreat in the study area. Not all the OC mobilized by RTSs is immediately transported to the nearshore zone; an important part is mobilized in the RTS floors. Therefore RTSs alter the OC budget of the nearshore zone by affecting the OC release process. Our results show that the contribution of RTSs to the nearshore OC budget is non-negligible and should be included when estimating the quantity of OC released from the Arctic coast to the ocean."*

3. I found the units of sediment and carbon counterintuitive and difficult to compare with other studies. Results are presented in absolute terms (total amount of carbon or sediment displaced from the whole study region) and it would be useful to state units normalized to area. Expressing material balance in terms of m2 would immediately let researchers unfamiliar with this area relate to the units and assess how important this process is. That would allow comparison of thermokarst mobilization of SOC and DOC to carbon release via active layer deepening. In this same vein, the number of features, which is focused on in the abstract and throughout the paper, seems immaterial compared to changes in area and volume. Ultimately, I had a hard time concluding at the end of the paper if thermo-erosion was increasing, decreasing, or remaining stable.

To make our statement more clear, we modified the units of material eroded and OC mobilized. For RTSs, we provide the stock estimates /per km of coast or /per RTS. For the RTSs that initiated after 1972 we provided estimates /per km of coast/yr or /per RTS/yr.
The reason why we initially showed the evolution of the number of features in Table 1 was because the increase in the number of RTSs explains the increase in coverage. RTSs increased in number but did not become larger.
However, to give a better overview to the reader and highlight our results better, we followed your advice and created a figure (Fig.3), combining the former Table 1 and 2.

4. It is unclear how/if uncertainties were propagated through this exercise. Absolute numbers are given, rather than ranges or estimates of center and standard deviation (e.g. all the tables and figures). Without measures of uncertainty, it is difficult to assess the reliability of these estimates or identify sources of that uncertainty in the analysis.

Uncertainties are indeed an important part to estimate. To improve our manuscript we added some description of the values we used from previous publications.

**Page 7, line 3:** "*To differentiate between the volumes of ice and sediments eroded, we used the volumetric ice content provided for each coastal segment in Couture and Pollard (2017). The model interpolates the data collected on 19 coastal segments to the whole Yukon Coast based on similarities between surficial geology and permafrost conditions. Ice contents were determined from shallow cores collected from upper soil layers and from bluff exposures.*"

**Page 7, line 19:** "*The OC values were derived from in-situ measurements collected at 31 locations and were interpolated to each coastal segment following the same approach as for the determination of ground ice (Couture, 2010). The SOC was measured for different soil unit layers along the bluffs and averaged for the upper first meter and lower meter of the soil columns (Couture, 2010). It therefore takes into account the heterogeneity of SOC contents at depth. DOC values account for the differences in DOC concentrations between wedge ice, massive ice and non-massive ice (Tanski et al., 2016), based on the ice volumes summarized in Couture and Pollard (2017). The OC values are therefore coarse but consistent for the whole Yukon Coast. The dataset is provided in supplementary material (S1_TableS1).*"

We also provided the range or standard deviations for the mean values of our results. We also show the variation in the dataset in the Figures 6 and 7.

5. There are multiple issues with visualizations particularly the stacked bar plots using a logarithmic y-axis and the reliance on tables. Stacked bar plots on a logarithmic scale are visually misleading since the ice volume, which represents the majority of material lost, appears negligible. Additionally, could the x-axis of these plots be organized by some salient ecological parameter (e.g. precipitation, climate, surficial geology) instead of by geographic position? This would help provide insight into processes driving these patterns. The use of tables is fine in some cases, but I wanted a figure showing rate of thermo-erosion (normalized by area) for the two time periods (1952-1972, 1972-2011), which seems like one of the key punchlines of this paper. The tabular form makes it harder to rapidly compare changes and trends and ultimately is not more compact than a (non-logarithmic) stacked barplot of those time periods.

We removed the former Tables 1 and 2 and replaced them by a Figure (Fig. 5), summarizing both tables and showing the changes in number and coverage of RTS per geologic unit and years.
We modified the Figures 6 and 7: we removed the logarithmic scale and created boxplots to give better estimates of the volumes of material eroded per RTS for each coastal segment. We added the geologic unit that underlies each segment in the x-axis.

6. To cryosphere scientists, the subject of this paper is immediately of interest, but I fear that the abstract and introduction do not provide enough context for a non-specialist to see the need and implications of the study. Defining key terms (e.g. active layer) and providing more context for why this process is of general interest would increase the impact of this paper.

We provide more information as well as defined terms such as active layer and retrogressive thaw slumps in the introduction.

7. The paper builds on many previous studies, but sometimes relies too heavily on explanations given in those studies. Especially on key issues like determining pre-formation ice content, DOC, and SOC, enough methodological detail should be given for

the reader to assess the approach. At the bare minimum, given that many of these estimates are highly uncertain (e.g. reconstructions of ice content), an explicit treatment of uncertainties and how uncertainties were propagated is necessary.

As mentioned above, we added these details in the Methods section:
**Page 7, line 3:** "*To differentiate between the volumes of ice and sediments eroded, we used the volumetric ice content provided for each coastal segment in Couture and Pollard (2017). The model interpolates the data collected on 19 coastal segments to the whole Yukon Coast based on similarities between surficial geology and permafrost conditions. Ice contents were determined from shallow cores collected from upper soil layers and from bluff exposures.*"

**Page 7, line 19:** "*The OC values were derived from in-situ measurements collected at 31 locations and were interpolated to each coastal segment following the same approach as for the determination of ground ice (Couture, 2010). The SOC was measured for different soil unit layers along the bluffs and averaged for the upper first meter and lower meter of the soil columns (Couture, 2010). It therefore takes into account the heterogeneity of SOC contents at depth. DOC values account for the differences in DOC concentrations between wedge ice, massive ice and non-massive ice (Tanski et al., 2016), based on the ice volumes summarized in Couture and Pollard (2017). The OC values are therefore coarse but consistent for the whole Yukon Coast. The dataset is provided in supplementary material (S1_TableS1).*"

**Line edits:**

**Page 1**
Line 10: An additional line introducing the general context would evaluable.
We added this sentence:
"*Retrogressive thaw slumps (RTSs) are among the most active thermokarst landforms in the Arctic and deliver a large amount of material to the Arctic Ocean. However, their contribution to the organic carbon (OC) budget is unknown.*"

Line 17-18: Standard SI format for number should be used (i.e. $8.6 \times 10^6$ not $8600 \times 10^3$). There are issues with this throughout the manuscript.
Modified throughout the manuscript.

Line 18: 53% of which was ice
Modified accordingly.

Line 21: 0.3% of the total OC flux for the Arctic Ocean? Unclear why this is of interest at this point in the paper. What percentage of the SOC stocks in the affected areas of the study region was mobilized by these features?
We adjusted this number to take into account the changes that we applied to our dataset and clarify this sentence as:
"*Between 1972 and 2011, 17% of the RTSs displaced $8.6*10^3$ m$^3$/yr of material, adding 0.6% to the OC flux released by coastal retreat along the Yukon Coast.*"

Line 25:I believe this estimate is for the entire permafrost zone, not just the Arctic
Modified accordingly:
**Page 1, line 26:**"*Soil organic carbon (SOC) stocks in the top three meters of soils, in deltas and the Yedoma regions across the northern circumpolar permafrost region are estimated to 1307 Pg; 76.4% (999 Pg) of them are stored in perennially frozen soils (Hugelius et al., 2014).*"

Line 27:Is it meant that air temperature has increased by approximately 3-4 degrees C? Air temperature in Celsius is expressed on a relative scale and it does not make senseto say increased by a factor of 3-4 (unless referring to change relative to absolute zero)
To take into account a new publication on Arctic warming, we replaced this sentence with:
**Page 1, line 29:***"Surface air temperature in the Arctic increased by 0.755°C per decade during 1998–2012 (Huang et al., 2017)."*

Line 31: Non-standard terminology for thermo-erosion features. Following Kokelj,Jorgenson, Fortier etc., thermo-erosion or thermal erosion are the blanket terms that include thermokarst (permafrost collapse) and other erosive processes associated with permafrost degradation.
We modified the sentence.
**Page 2, line 10:** *"Thermokarst and thermo-erosional processes occur by the thawing of ice-rich permafrost and the melting of massive ice."*

**Page 2**
Line 5: Consider including more recent modeling studies such as Koven et al.2015, Kessler 2017, or Sudakov and Vakulenko
Thank you for this suggestion.
Koven et al. (2015) published estimations of permafrost thaw feedback based on the distribution of carbon in the soils. Kessler (2017) measured the economic coast of the permafrost carbon feedback. Sudakov and Vakulenko (2014), developed a mathematical model to constrain the permafrost carbon feedback using methane emission data.
In this part of the introduction we mention the impact of carbon stocks on the global greenhouse gas emission. We therefore added the references to the 2 first publications in our introduction.
**Page 1, line 34:** *"Permafrost carbon stocks were only recently included in calibrating global carbon models, highlighting a relevant contribution of thawing permafrost to the overall climate and economic response to human greenhouse gas emissions (Kessler, 2017; Koven et al., 2015; MacDougall et al., 2012; Burke et al., 2012; von Deimling et al., 2012)."*

Line 9: Consider citing Abbott et al. 2016 or McGuire et al. 2016, which summarize current modeling uncertainties stemming from exclusion of these parameters. Both of these studies directly support the need for the current study by emphasizing the importance of constraining thermo-erosion.
We added a sentence in the text to take into account the conclusions of both studies:
**Page 2, line 7:** *"Both expert assessments (Abbott et al. 2016) and model evaluations (McGuire et al., 2016) identified permafrost degradation as one of the most important sources of uncertainty in predicting the timing and magnitude of the permafrost carbon feedback."*

Line 25: Word choice (potentially control or influence rather than forcing)
Modified accordingly: "control".

Figure 1: Really nice figure. Potentially put the specific reach names in the SI (not of interest to most readers)
Modified accordingly.

**Page 6**
Line 9: "In order to" can always be replaced by "To"
Modified accordingly in the whole manuscript.

Line 8: How was uncertainty for the compound assumptions in these analyses dealt with? Need more detail generally.
To clarify this point, we described the data we used from other publication with more details:
**Page 7, line 3:** "*To differentiate between the volumes of ice and sediments eroded, we used the volumetric ice content provided for each coastal segment in Couture and Pollard (2017). The model interpolates the data collected on 19 coastal segments to the whole Yukon Coast based on*

*similarities between surficial geology and permafrost conditions. Ice contents were determined from shallow cores collected from upper soil layers and from bluff exposures."*

**Page 7, line 19:** "*The OC values were derived from in-situ measurements collected at 31 locations and were interpolated to each coastal segment following the same approach as for the determination of ground ice (Couture, 2010). The SOC was measured for different soil unit layers along the bluffs and averaged for the upper first meter and lower meter of the soil columns (Couture, 2010). It therefore takes into account the heterogeneity of SOC contents at depth. DOC values account for the differences in DOC concentrations between wedge ice, massive ice and non-massive ice (Tanski et al., 2016), based on the ice volumes summarized in Couture and Pollard (2017). The OC values are therefore coarse but consistent for the whole Yukon Coast. The dataset is provided in supplementary material (S1_TableS1).*"

Line 12: Why were these processes not included? How does that affect the estimates?
In the first manuscript we decided to leave aside these two processes because we did not have accurate erosion rates for the area. Meanwhile, a new study from our colleagues was accepted for publication in JGR:Earth Surface (Irrgang et al., 2017, in review). We therefore modified our dataset to take into account:
   1. the 5.5% of material that subside in the slump floors (Obu et al., 2016)
   2. the area of the slump that is being washed away by coastal retreat yearly. For this we used coastal rates of change from the study from Irrgang et al., 2017
We added the previously Figure 6 in the method section as Figure 4 and clarified out methodology:

*"**3.2.3 Volume of eroded material**
To calculate the volume of eroded material from the headwall of the RTS identified in 2011, we subtracted the mean surface elevation values obtained from the LiDAR dataset from the mean interpolated surface elevation values (Fig. 3). However, these volumes do not account for the material eroded from the RTS headwalls that settles within the RTS floors and for the material eroded and transported alongshore by coastal retreat (Fig. 4). Due to ground ice melting, ca. 5.5% of the reworked sediments subside and remain compacted in the RTS floor (Obu et al., 2016). We therefore adjusted the material volumes based on this value (Fig.4, c). Additionally, we measured the volumes of material eroded and transported by coastal retreat using the rate of coastal change between 1952 and 2011 from Irrgang et al. (2017). Using this rate, we calculated the volumes of eroded material between 1952 and 2011 for each RTS. For the RTSs that initiated after 1972, calculated the volumes of eroded material between 1972 and 2011 (Fig. 4, d).
To differentiate between the volumes of ice and sediments eroded, we used the volumetric ice content provided for each coastal segment in Couture and Pollard (2017). The model interpolates the data collected on 19 coastal segments to the whole Yukon Coast based on similarities between surficial geology and permafrost conditions. Ice contents were determined from shallow cores collected from upper soil layers and from bluff exposures.*

[Figure]

*Figure 4: Cross-section of a retrogressive thaw slump (RTS) illustrating the calculated and omitted volumes of sediments eroded through slumping between 1972 and 2011. The calculation estimates the amount of material released to the nearshore zone through slumping (b) and takes into account the material eroded from the RTS headwalls that remains within the RTS floors where it settles (c), and (d) the material eroded and transported alongshore by coastal erosion. The volumes of material that remains within the RTS floors were estimated from Obu et al. (2016)."*

**Page 7**

Line 6-26: With the presented information, it is not clear if these estimates were downscaled from measurements of fluxes at feature outlets or if they are inferred from the mass of SOC there previously multiplied by volume displaced. If the latter, how are vertical differences in SOC accounted for this this framework?

We clarified our methodology at the beginning of the section 3.3:

**Page 7, line 17:** *"We inferred mobilized SOC and DOC stocks and fluxes from RTSs from the mass of SOC and DOC per meter column in each coastal segment provided in Couture (2010) and Tanski et al. (2016) in relation to the estimated volume of material displaced by each RTS. The OC values were derived from in-situ measurements collected at 31 locations and were interpolated to each coastal segment following the same approach as for the determination of ground ice (Couture, 2010). The SOC was measured for different soil unit layers along the bluffs and averaged for the upper first meter and lower meter of the soil columns (Couture, 2010). It therefore takes into account the heterogeneity of SOC contents at depth. DOC values account for the differences in DOC concentrations between wedge ice, massive ice and non-massive ice (Tanski et al., 2016), based on the ice volumes summarized in Couture and Pollard (2017). The OC values are therefore coarse but consistent for the whole Yukon Coast. The dataset is provided in supplementary material (S1_TableS1).*

**Page 8**

Line 3: Focusing on the number of features doesn't seem terribly relevant to the question of the permafrost climate feedback. The area and volume results are more informative. In general, a few clear figures would more effectively communicate the observed patterns.

It is certain that the change analysis in area and volume of RTSs is more relevant to the question of the permafrost climate feedback. However, we show that the increase in RTS coverage (14%), is mostly driven by an increase in the number of RTSs more than y a growth in the size of the RTSs. This is the reason why we decided to emphasize that there is an increase in single RTS features along the coast, which causes an increase in the total coverage of RTSs.

We removed the Tables 1 and 2 and added a figure (Fig. 5) to better visualize both increases through time: in number and total coverage of RTSs.

Table 1: This would be more compelling in figure form. If table is retained, no need to use cryptic acronyms in the first column (i.e. L, Mm, Mr). There is enough room to spell out the parameters
This information is now displayed in the Figure 5.

**Page 9**
Table 2 would also be more effective in figure format. As currently presented, it is hard to tease apart what is changing across the time series.
This information is now displayed in the Figure 5.

**Page 10**
Table 3: This should be normalized to area covered by the geologic units. Are some of the units displacing more material per unit area or are the differences due to different relative coverage? No estimates of uncertainty are given.
We normalized the values according to the coastal length of the geologic units in the study area. We chose to normalize the values by the coastal length in km because we only mapped coastal RTSs. We now show volumes in $m^3$ / km. See Table 3 and 4.

Figures 4 and 5. Problematic to show a stacked bar plot with a logarithmic axis.
We modified the Figures 4 and 5 to take into account this comment. We removed the logarithmic axis and showed boxplots providing information on the material released for each coastal segment.

**Page 14**
Line 18: Good example of why estimates should be normalized by area (i.e. expressed on a kg m2 yr basis or in mm/yr for sediment)
Following this comment, we decided to present the results as $m^3$/yr, $m^3$/ RTS and $m^3$/km of coast in the whole manuscript. We could not calculate fluxes for the 162 RTSs as we do not know when they initiated.

Line 27: Still not clear how this affects the analysis. If the question is about total sediment and carbon balance, these processes, which are caused or at least facilitated by thermo-erosion seem pertinent
As explained above, we took into account these processes and therefore modified the section 5.2 of the manuscript.
*"5.2 Eroded material from RTSs and OC fluxes*
*The expansion of RTSs along the coast causes the displacement of large volumes of material from the land to the sea. We show that 56% of the RTSs identified in 2011 (162 RTSs) have reworked at least 16.6\*106 m3 of material along the Yukon Coast, which is 102.5\*103 m3/RTS of material eroded per RTS. Among these RTSs, 49 RTSs initiated after 1972 and reworked 27.2\*103 m3/yr of material, which is 0.6\*103 m3/RTS/yr . These estimates are low compared to material removal from other RTSs in the Arctic. Lantuit and Pollard (2005) calculated a sediment volume loss of 105\*103 m3 between 1970 and 2004 for a single RTS located on Herschel Island; Kokelj et al. (2015) and Jensen et al. (2014) measured material displacements up to 106 m3 per RTS located in NW Canada and Alaska; the Batagay mega-slump located in Siberia eroded more than 24\*106 m3 of ice rich permafrost in 2014 (Günther et al., 2015). The size of the observed RTSs is one reason behind such differences: most of the RTSs examined in the above studies are classified as mega slumps (> 0.5 ha). The RTS studied in Lantuit and Pollard (2005) was the largest RTS identified along the entire Yukon Coast in 2011, 24 ha. However most of RTSs along the Yukon coast are small, with an average size of 0.2 ha (Ramage et al., 2017). This has implication for studies that attempt to model the impact of RTSs on the eroded material budgets in the Arctic.*
*Couture (2010) estimated the annual flux of mineral sediment eroded by coastal retreat along the Yukon Coast to 7.3\*106 kg/km/yr. We show that along a 190-km portion of the Yukon Coast, 17% of the RTSs identified along the coast in 2011 (49 RTSs) contributed to 1% of the annual flux of material eroded along the Yukon Coast (61\*103 kg/km/yr). These RTSs initiated after 1972 incised 1% (2 km) of the coastline in 2011 and were on average smaller than the average RTSs.*

*Increasing the number and areal coverage of coastal RTSs has therefore large consequences on the flux of eroded material along the Arctic coasts.*
*We estimated the annual OC fluxes (SOC and DOC) from these 49 RTSs to 1.3\*103 kg/km/yr, including 0.02 kg/km/yr DOC. The average OC flux from coastal retreat along the entire Yukon Coast is 157\*103 kg/km/yr (Couture, 2010) with an average DOC flux of 0.2\*103 kg/km/yr (Tanski et al., 2016). We show that the annual OC flux released by the 49 RTSs initiated after 1972 was 0.6% the annual OC flux from coastal retreat. Most of these fluxes originated from ice thrust moraines, where the number of RTS initiated after 1972 was the highest. RTSs develop mainly on ice-thrust moraines because of the presence of large volumes of massive ground ice (Ramage et al., 2017). As a result, only half of the material eroding from the RTS headwall is sediment and most of the OC is released as DOC."*

**Page 16**
Line 20: I think the authors are referring to Abbott and Jones 2015
Modified accordingly.

---

## Author Comment (AC2) · 18 Jan 2018

Thank you for the thorough revision of our manuscript and your constructive comments that helped to improve the paper. Our replies to your comments are combined in the .pdf "Answer_reviewer2".

Further modifications can be found in the reviewed version of the manuscript.

Best, Justine Ramage

Please also note the supplement to this comment:

[Figure]

https://www.biogeosciences-discuss.net/bg-2017-437/bg-2017-437-AC2-supplement.pdf

**Supplement:**

*We thank the two reviewers and the editor for the thorough revision of our manuscript and their constructive comments that helped to improve the paper. Our replies to the comments are written in green. Line numbers given in our replies refer to the revised version of the manuscript.*

**Anonymous Referee #2**

The submitted paper 'Contribution of Coastal Retrogressive Thaw Slumps to the Nearshore Organic Carbon Budget along the Yukon Coast' by Ramage et al. gives an indication of the specific impacts of slumps on the sediment budget and on the carbon budgets of Arctic tundra coasts in northern Canada. The focus is on three main topics as stated at the end of the introduction: 1) definition, quantification and temporal analysis of RTSs; 2) estimation of sediment/ice and OC budgets related to these slumps; and 3) measure the OC fluxes between 1972 and 2011. Looking to these aims of the study, I have some specific comments related to these different goals and will come with suggestions to restructure some parts of the paper to make it more focused on the RTSs. The data presented is very valuable and the paper will after restructuring be a valuable contribution to better understand Arctic coastal environments and its changes.

**Ad 1)** The coastal stretch in NW Canada is probably very representative for a large part of the Arctic coastal environments. The different geomorphological units (or geological units as stated in Figure 1) cover a wide range of environments and the coastal stretch with its units can probably be used to upscale the findings. You can state explicitly that you use the findings along this stretch to upscale in the future.
We are not sure we understand correctly this comment.
Ground ice and OC data from Couture (2010), Couture and Pollard (2017) and Tanski et al. (2016) were upscaled from single field sites to the entire coastal segments. These data were further interpolated to costal segments showing similar permafrost conditions.
We added more details in the methods sections 5.2 and 5.3.

We did not upscale our data in the future. The number of RTS and their size were mapped based on 3 type of imagery from 1952, 1972 and 2011.
Volumes of material and OC stocks were estimated for a subset of the number of RTSs identified in 2011. OC fluxes were calculated for RTSs that initiated after 1972.

Upscaling our results to the future would require a more complex approach, including changes in temperature, precipitations and sea ice properties, which are major controls for RTSs development.

Use Figures 2 and 6 to define what RTSs are.
We added a definition of RTSs in the introduction and described them further in the methods section.
**Page 2, line 12:** *"Retrogressive thaw slumps (RTSs), a type of slope failure caused by permafrost thaw, (...)"*

Tables 1 and 2 give an indication of the amount and sizes of RTSs for different units. It would be good to start the explanation of the spatial pattern, followed by the development in time. Now, it starts with changes in time, without having an idea how many and how large the RTSs are in the different units.
We described the current (2011) distribution of RTSs in a paper published in 2017: Ramage, J.L., Irrgang A.M, Herzschuh U., Morgenstern A., Couture N., Lantuit H.: Terrain Controls on the Occurrence of Coastal Retrogressive Thaw Slumps along the Yukon Coast, Canada. *Journal of Geophysical Research: Earth Surface,* 2017.

We replaced the tables 1 and 2 by a Figure (Fig.5), combining the information provided in both Tables.

Another thing is the use of the terms active or stable RTS. What kind of conditions do you use to call a RTS active or stable? Is it related to fresh scarps, vegetation coverage?
To clarify this information to the reader, we added this information in the section 3.1:
**Page 4, line 7:** *"Active RTSs are characterized by steep headwalls exposing ice-rich permafrost, slump floors with thawed sediments, and incised gullies. Stable RTSs comprise gently sloping and vegetated headwalls, vegetated slump floors, and no visible active gully systems (Ramage et al., 2017; Lantuit and Pollard, 2008; Wolfe et al., 2001)."*

**Ad 2)** Estimation of the sediment/ice erosion due to RTS and OC budgets is quite straightforward and the best you can do with the limited Lidar data. Figure 6 in your discussion is very nice and can be used to explain your estimation of budgets in an earlier phase.
Thank you for the suggestion. We move the Figure 6 to the method section and renamed it as Figure 4 and clarified the methodology:

*"**3.2.3 Volume of eroded material***
*To calculate the volume of eroded material from the headwall of the RTS identified in 2011, we subtracted the mean surface elevation values obtained from the LiDAR dataset from the mean interpolated surface elevation values (Fig. 3). However, these volumes do not account for the material eroded from the RTS headwalls that settles within the RTS floors and for the material eroded and transported alongshore by coastal retreat (Fig. 4). Due to ground ice melting, ca. 5.5% of the reworked sediments subside and remain compacted in the RTS floor (Obu et al., 2016). We therefore adjusted the material volumes based on this value (Fig.4, c). Additionally, we measured the volumes of material eroded and transported by coastal retreat using the rate of coastal change between 1952 and 2011 from Irrgang et al. (2017). Using this rate, we calculated the volumes of eroded material between 1952 and 2011 for each RTS. For the RTSs that initiated after 1972, calculated the volumes of eroded material between 1972 and 2011 (Fig. 4, d).*
*To differentiate between the volumes of ice and sediments eroded, we used the volumetric ice content provided for each coastal segment in Couture and Pollard (2017). The model interpolates the data collected on 19 coastal segments to the whole Yukon Coast based on similarities between surficial geology and permafrost conditions. Ice contents were determined from shallow cores collected from upper soil layers and from bluff exposures.*

[Figure]

*Figure 4: Cross-section of a retrogressive thaw slump (RTS) illustrating the calculated and omitted volumes of sediments eroded through slumping between 1972 and 2011. The calculation estimates the amount of material released to the nearshore zone through slumping (b) and takes*

*into account the material eroded from the RTS headwalls that remains within the RTS floors where it settles (c), and (d) the material eroded and transported alongshore by coastal erosion. The volumes of material that remains within the RTS floors were estimated from Obu et al. (2016)."*

The presentation in Figures 4 and 5 is often a bit confusing. You followed a spatial axis on the x-axis, but you don't use this in the rest of the discussion. It would probably more interesting to group it according to the geomorphologic/geologic unit and discuss this variation in time. You also gave this unit related results in tables 3,4 and 5. The volumes of eroded materials will be better visible if you don't use a cumulative sediment and ice volume on a logarithmic Yaxis, but come with values for sediment and ice (different symbols).
We modified the Figures 6 and 7. We removed the logarithmic axis and plotted the material released per RTS for each coastal segment. We did not group the values by geologic unit because we present those results in the tables 1 and 2.

**Ad 3)** I wonder why you made a separate aim only following OC budgets in time. I think it is more logical to describe the estimation of budget terms under 2) and thereafter discuss all terms (sediment / ice / OC) in more detail in time.

In the section 4.2, we use 2 different dataset:
1. 162 RTSs that were identified on the 2011 imagery. We don't know when those RTSs were initiated
2. 49 RTSs that were identified on the 2011 imagery but not on the 1972 imagery. We named those RTSs "RTSs initiated after 1972". For these we were able to calculate fluxes.

We initially described the volumes of eroded material for both of these dataset and then described the different parts (sediment / ice / OC), as you suggested. However, we found difficult for the reader to distinguish between the 2 datasets and decided to talk first about the volumes of material and related stocks of sediment, ice and OC and then to focus more on the fluxes using the dataset 2.

**Discussion and conclusions**
The discussion of the paper is now in 4 subsections (erroneous numbered 5.1, 5.2, 5.3 and 5.3). Following the three aims and the structure of the results, it is perhaps very attractive to start a discussion about the 'static' description of RTSs (sizes, amounts, coupling to geo units) and about the uncertainties in determining the RTSs using the data set. This can be followed by a second section about the changes in slump activity (your acceleration of slump activity). Then we have two sections related to the second aim: your sections about Eroded material from RTSs and Calculated OC fluxes. Finally, you can place it in a broader prespective as you tried to do in 5.4. This can also include some remarks about upscaling to Arctic shorelines.
Thank you for these suggestions. We modified the discussion following the points 2,3 and 4. The "static" description of RTSs for 2011 was already done in Ramage et al. (2017). We added a sentence at the beginning of the section 5.1 to mention the results of this previous study.

We kept the first section of the discussion 5.1 on the evolution of RTSs along the coast. We then merged the previous sections 5.2 and 5.3 into a section 5.2 on the Eroded material from RTSs and calculated OC fluxes and kept section 5.4 renamed as 5.3.

**The conclusions** are to the point.

**Title** The title is not covering the work done. You have showed many more results on the losses of ice and sediments and its changes in time as well. Impact not only OC budgets.
Following your suggestion, we modified the title of the manuscript.

**Line edits:**

**Page 2**
line 9: Hugelius et al., 2014 is not in the reference list.
Thank you for noticing. We added the citation in the reference list.

**Page 5**
line 20: n=125 refers to?
The n = 125 refers to the number of RTSs discarded. We modified the sentence to
*"We discarded the 125 RTSs outside of the LiDAR scan from the volume and flux analyses."*

**Page 6**
Figure 3: Are all splines in the RTSs giving a sloping surface from N to S?
Not all splines give a looping surface from N to S in the study area. The orientation of the splines depends on the orientation and the topography of the coast surrounding the RTSs on which RTSs occur. On the Figure 3, all RTSs are facing south because the example is taken from a south facing and sloping coastline. We decided to modify Figure 3 in order to clarify this point.

line 12: Can you estimate the coastal retreat impact (or assume ..%)?
In the first manuscript we decided to leave aside these two processes because we did not have accurate erosion rates for the area. Meanwhile, a new study from our colleagues was accepted for publication in JGR:Earth Surface (Irrgang et al., 2017, in review). We therefore modified our dataset to take into account:
1. the 5.5% of material that subside in the slump floors (Obu et al., 2016)
2. the area of the slump that is being washed away by coastal retreat yearly. For this we used coastal rates of change from the study from Irrgang et al., 2017

We added the previously Figure 6 in the method section as Figure 4 and clarified out methodology (section 5.2.2 and 5.2.3).

**Page 11**
Figure 4: Only Mr?
Thank you for pointing out the mistake. Following the recommendations of the other reviewer, we modified the figure.

**Page 13**
line 21: Wolfe and Dallimore or Wolfe et al. (see reference list)
We modified the reference as Wolfe et al., 2001.

**Page 16**
line 13: Should be section 5.4.
Changed accordingly.